# Molecular Developmental Biology of Fibrodysplasia Ossificans Progressiva: Measuring the Giant by Its Toe

**DOI:** 10.3390/biom14081009

**Published:** 2024-08-15

**Authors:** O. Will Towler, Eileen M. Shore, Frederick S. Kaplan

**Affiliations:** 1Division of Plastic Surgery, Children’s Hospital of Philadelphia, Philadelphia, PA 19104, USA; towlero@chop.edu; 2Center for Research in FOP and Related Disorders, Perelman School of Medicine, University of Pennsylvania, Philadelphia, PA 19104, USA; shore@pennmedicine.upenn.edu; 3Department of Orthopaedic Surgery, Perelman School of Medicine, University of Pennsylvania, Philadelphia, PA 19104, USA; 4Department of Genetics, Perelman School of Medicine, University of Pennsylvania, Philadelphia, PA 19104, USA; 5Department of Medicine, Perelman School of Medicine, University of Pennsylvania, Philadelphia, PA 19104, USA

**Keywords:** FOP, ACVR1, skeletal development, joint development, BMP, digit patterning

## Abstract

When a genetic disease is characterized by the abnormal activation of normal molecular pathways and cellular events, it is illuminating to critically examine the places and times of these activities both in health and disease. Therefore, because heterotopic ossification (HO) in fibrodysplasia ossificans progressiva (FOP) is by far the disease’s most prominent symptom, attention is also directed toward the pathways and processes of bone formation during skeletal development. FOP is recognizable by effects of the causative mutation on skeletal development even before HO manifests, specifically in the malformation of the great toes. This signature skeletal phenotype is the most highly penetrant, but is only one among several skeletal abnormalities associated with FOP. Patients may present clinically with joint malformation and ankylosis, particularly in the cervical spine and costovertebral joints, as well as characteristic facial features and a litany of less common, non-skeletal symptoms, all stemming from missense mutations in the *ACVR1* gene. In the same way that studying the genetic cause of HO advanced our understanding of HO initiation and progression, insight into the roles of ACVR1 signaling during tissue development, particularly in the musculoskeletal system, can be gained from examining altered skeletal development in individuals with FOP. This review will detail what is known about the molecular mechanisms of developmental phenotypes in FOP and the early role of ACVR1 in skeletal patterning and growth, as well as highlight how better understanding these processes may serve to advance patient care, assessments of patient outcomes, and the fields of bone and joint biology.

## 1. Introduction

Fibrodysplasia ossificans progressiva (FOP; MIM#135100) is a genetic disease characterized by the progressive immobilization of the body due to heterotopic ossification (HO) that restricts the function of joints [1,2]. Although HO in FOP is not detected prenatally, after birth, HO typically forms following the activation of the immune system through injury, vaccination, or illness, or spontaneously in an event of high, localized immune activity called a flare-up [3,4]. HO in FOP occurs through endochondral ossification (EO), similar to that of bone formation in most of the developing embryonic skeleton. It is distinct, however, in that EO during early development is tightly regulated and not associated with immune cell activity or inflammation, whereas HO grows irregularly and is commonly associated with both immune activity and inflammation [5,6,7]. Mature HO is distinct from normal bone in its aberrant morphology and mineral density, but histological examination reveals chondrocytes, osteoblasts, osteocytes, calcified tissue, and vasculogenesis—all hallmarks of healthy bone [5]. When expanding HO abuts primary skeletal structures, it may even form pseudo-joints, or systems in which dense tissue in the HO mass articulates with the skeleton and causes a mobile, painful, joint-like system. Together, these features support the assertion that the HO of FOP recapitulates many, but not all, of the developmental processes of EO in the skeleton, despite being distinct in its causation, timing, and location.

FOP has long been associated with characteristic skeletal malformations, particularly those of the great toe [8]. Therefore, we look to the endogenous skeleton for clues as to how the normal processes and pathways of bone formation could be co-opted elsewhere in the body by FOP. Skeletal maturation is dependent on bone morphogenetic proteins (BMPs), which were first identified and named for the ability to induce the formation of ectopic bone [9]. To regulate its potent morphogenetic ability, BMP pathway signaling is carefully coordinated among myriad extracellular and intracellular activators and inhibitors. Because of this bone-forming capacity, components of the BMP signaling pathway were among the first targets queried in molecular studies of the pathogenesis of FOP [10]. Among these is the receptor culpable for FOP, activin A receptor type I (ACVR1), also called activin receptor-like kinase-2 (ALK2) [11]. ACVR1 is a type I BMP receptor that, in unaffected individuals, helps regulate the initiation of an intracellular cascade to activate the genetic program of EO [12]. At sites of BMP pathway activity during embryonic skeletal formation, skeletal progenitor cells condense and differentiate into chondroblasts [13]. These chondroblasts mature, adjust their extracellular environment, and recruit additional cells to complete the metamorphosis of soft tissue into mineralized, vascularized bone. Therefore, the location and timing of BMP activity in the presence of cells capable of responding to it is of paramount importance. Otherwise, EO would proceed directionless and unrestricted, preventing the formation of the joints, tendons, and ligaments that allow the skeleton to move.

This review will detail the molecular, cellular, and tissue-level events of skeletal morphogenesis and how these are disrupted in the early development of individuals with FOP. It will subsequently discuss post-natal manifestations of enhanced BMP signaling through the mutant ACVR1 receptor. Finally, we will examine the impact of these phenotypes on patients and what that may mean for the discovery and assessment of current and future therapies for treating this devastating disease.

## 2. Molecular Basis of FOP

To understand how FOP-causing mutations disrupt signaling in various contexts, we will first review the BMP signaling pathway and its role in the endochondral ossification of the long bones of the limbs. Finer details of BMP pathways and the broader TGF-ß superfamily have been comprehensively reviewed elsewhere [14,15,16].

The initiation of canonical BMP pathway signaling requires the formation of a heterotetrameric complex composed of two type I and two type II transmembrane BMP receptors. These bind an extracellular dimer of two BMP ligands [17]. The formation of this complex promotes receptor conformational changes inside the cell to destabilize binding with the intracellular inhibitory protein FKBP12 and allow the serine/threonine kinase domain of the type II receptors to phosphorylate the glycine/serine-rich (GS) domain of the type I receptors [18]. This, in turn, phosphorylates the intracellular signal transducer complex, SMAD1/5/(8/9), which prompts the recruitment of the chaperone protein co-SMAD4 and the translocation of the active SMAD complex to the nucleus. This signal transduction via pSMAD1/5/(8/9) is called the canonical BMP pathway. Once it is in the nucleus, binding to BMP-responsive elements (BMEs) drives the transcription of target genes, including *SOX9*, a master regulator of chondrogenesis [19]. BMP signaling complexes may also phosphorylate TAK1 to activate the non-canonical p38 mitogen-activated protein kinase (MAPK) pathway and drive the transcription of *RUNX2*, *DLX5*, and *SP7*, leading to osteogenesis [15].

The assembly of various ligands and receptors into this membrane-bound complex and the subsequent selection of intracellular transducers—whether pSMAD1/5/(8/9) for the canonical BMP pathway, TAK1 for the non-canonical BMP pathway, or pSMAD2/3 for the TGFß pathway—is what permits the precise spatiotemporal modulation of BMP pathway activation [20,21,22]. Complexes containing overlapping constituent proteins may activate or inhibit the BMP pathway or activate the TGF-ß pathway [22]. Because these pathways are multimodal and present in a vast variety of developmental contexts, the absence or mutation of individual members may yield broad or specific consequences.

We consider these functions now in the context of the normal development of long bones, such as those of the limbs. In this process, canonical BMP pathway activation in condensations of mesenchymal cells causes endochondral ossification in embryonic tissues [13,23,24]. Between these condensations, BMP signaling is inhibited by Noggin, which permits the development of joint tissues [25]. These condensations undergo chondrogenesis in a stepwise fashion. First, BMP signaling and the subsequent expression of its direct transcriptional target *SOX9* begin a transcriptional cascade, driving the differentiation of these cells into chondroblasts [19]. Chondroblasts proliferate and become polarized according to local directional molecular and physical cues. As these nascent chondroblasts stiffen the matrix around them with collagen, increasingly hypoxic conditions cue vasculogenesis through HIF1α activity and VEGF signaling. BMP signaling causes chondrocytes in the center of the cell mass to undergo hypertrophy and apoptosis or transdifferentiate to osteoblasts, which begin mineralizing the matrix around them. The newly stiffened and mineralized ECM plays a critical role in mesenchymal cell fate determination, with stiffer substrates promoting differentiation into osteoblasts and osteocytes [26].

As these bone anlagen progressively elongate, cells near the ends become organized in distinct layers. BMP is tightly controlled in a zonal fashion among these layers to allow other signals, including Wnt, PTHRP, and IHH, to maintain layers of articular cartilage at the epiphyses of the bone and layers of proliferating cells in the growth plates, which permit smoothly articulating joints and the longitudinal growth of bones, respectively [20,27]. The disruption of these balanced signals can diminish bone growth, lead to the premature loss and closure of growth plates, and prevent the formation and maintenance of articular cartilage [28,29]. Thus, healthy bone development is dependent on the regulation of BMP signaling, mechanotransduction, and hypoxia, all of which are implicated in models of FOP [30,31,32].

The processes and pathways described above are critically disrupted by FOP-causing mutations in the *ACVR1* gene. The ACVR1 receptor is a type I BMP receptor typically found in canonical BMP-pathway-activating complexes [33,34]. It partners with other type I receptors, either BMPR1A/ALK3 or BMPR1B/ALK6 [35], and the type II receptors BMPR2 and ACVR2a [36,37]. It is the preferred target of BMP6 and BMP7 ligands and is also activated by BMP2 and BMP4 [17]. Most FOP patients share the same ACVR1 mutation: the replacement of arginine with histidine at position 206 (ACVR1-R206H) [11]. This and several other rarer pathogenic mutations cluster either within the GS domain, near the FKBP12-binding region, or within the protein kinase domain [38,39]. These non-R206H mutations are referred to as “variant” cases. Crystallography suggests R206 and Q207 stabilize FKBP12 binding to the GS domain [40] and that the R206H mutation reduces the stability of the FKBP12-ACVR1 complex compared to wild-type protein, which permits “leaky” signaling [41]. Experiments in mammalian cells show a reduced requirement of the BMP ligand for canonical pathway activation by ACVR1-R206H and the hyperactivation of the pathway in the presence of ligands [30]. The mutant receptor also shows reduced responsiveness to canonical inhibitors of signaling such as Noggin [42,43]. Experiments in zebrafish show a reduced requirement for partner type I receptors and the ligand binding of both classic and variant FOP receptors [44]. This reduced receptor requirement is accompanied by changes in GS domain activation requirements, permitting ACVR1-R206H to activate the SMAD complex more readily [44,45]. Together, these data show that ACVR1-R206H effectively raises the basal rate of BMP pathway activation and bypasses regulation by usual molecular mechanisms.

FOP-causing mutations also alter the ACVR1 receptor’s response to BMP/TGFß family ligands. BMP2, 4, 6, and 7 typically strongly activate the canonical BMP signaling pathway through ACVR1. In cells expressing either the recurrent R206H mutation or one of several variant FOP-causing mutations, BMP4 treatment yields a much higher phosphorylation of SMAD1/5/(8/9) [46]. Additional cell culture experiments revealed the enhanced responsiveness of ACVR1-R206H to BMP2, 4, 6, and 7 in transfected C2C12 cells [42], and to BMP6 and 7 in patient-derived induced pluripotent stem cells [37]. The mutations can also change context-dependent functions of certain ligands. Activin A binding to ACVR1-containing tetramers can act as a BMP pathway inhibitor by locking receptors into non-signaling complexes [47,48]. However, in the presence of the ACVR1-R206H mutation, Activin A strongly promotes BMP pathway activity, and blocking Activin antibodies in the presence of the FOP mutation decreases the incidence of trauma-induced HO [48,49,50,51,52]. The suppression of HO formation by palovarotene, an RARγ agonist, in individuals with FOP is accompanied by a reduced expression of *Inhba*, the gene encoding Activin A, further supporting a role for Activin A in the molecular pathogenesis of FOP [53].

The canonical transducer of BMP signaling, pSMAD1/5/(8/9), is shared with the RhoA pathway, which is a mechanotransduction pathway important for regulating cells’ movements and how they sense their environments [54,55]. By aberrantly phosphorylating SMAD, FOP mutations alter cells’ ability to sense and respond to physical forces in the microenvironment due to the crosstalk between these two pathways [31,56]. This increases the nuclear translocation of the mechanosensing pathway proteins YAP and TAZ, the expression of cartilage- and bone-associated genes, and the consequent differentiation of mesenchymal cells towards cartilage and bone fates [31,56].

In hypoxic conditions, BMP signaling complexes are retained in active states in endosomes. Cells with ACVR1-R206H mutant receptors retain these hypoxia-dependent endosomes longer than those with wild-type ACVR1, thereby extending the duration of enhanced BMP signaling [32]. Because bone-forming environments tend to be hypoxic and because the inhibition of HIF1α, a hypoxia-sensing protein, abrogates HO formation in models of FOP, HIF1α is a potent target for FOP therapies. Furthermore, the mechanistic target of rapamycin (mTOR) acts upstream of HIF1α. Targeting mTOR has been successful in human cell models and mouse models in reducing HO [57,58] and is an important signaling molecule in joint disease and maintenance, making it a promising candidate for tackling both HO and the non-HO symptoms of FOP. These two factors have recently been reviewed elsewhere [59].

## 3. The Great Toe of FOP

One of the most curious features of FOP is also one of the most important for providing insight into how the disease affects skeletal biology and the role of ACVR1 in early development: the bilateral malformation of the great toes [8]. At birth, most individuals with FOP are noted to have unusual, laterally deviated great toes. This characteristic FOP malformation is typically referred to as hallux valgus with absent or malformed joints. All affected individuals have malformation of the distal metatarsal, while half of cases show a loss of one phalanx, and approximately one-third show a longitudinal epiphyseal bracket (a condition almost never observed in the hallux except in FOP), in which the growth plate extends longitudinally along the phalanx [60]. FOP variant mutations may lead to much more severe digit phenotypes, including syndactyly and severe reduction in multiple digits in both the hands and feet [38]. These together suggest a disruption in the proximal–distal pre-patterning of the phalanges and begin to paint a picture of the molecular pathogenesis of the development of the normotopic skeleton of FOP.

In normal vertebrate development, digits of the hands and feet are patterned from proximal to distal according to a pattern of BMP pathway activity in which high activity yields a phalanx, metacarpal, or metatarsal, and low activity denotes a joint interzone (Figure 1 [21,61]. In humans, mice, and most related mammals, this pattern yields three phalanges in each of digits 2–5, and two phalanges in digit 1, which corresponds to both the thumb and the great toe. As with other long bones of the limbs, phalanges begin as primary ossification centers that grow circumferentially while lengthening along the proximo-distal axis, aided by growth plates containing rapidly dividing chondroprogenitor cells [62]. Physical or molecular disruptions may alter the pattern along this axis, thereby changing the number of joints or skeletal elements, or alter the direction of growth, leading to bunions or other altered morphology.

Great toe malformation is represented in genetic mouse models of FOP [63,64], providing a unique opportunity to study the etiology of this phenomenon. However, *Acvr1* is necessary to complete gastrulation in mice, and the global expression of the R206H mutation in mice leads to perinatal lethality [64,65,66], necessitating that experimental systems restrict knock-out and knock-in alleles to specific cell lineages. *Acvr1* is expressed at low levels during early murine skeletogenesis and is rapidly restricted to expression in the periosteum, the cell layer surrounding the hardened cortical bone in long bones like those of the limbs [12,67]. Mice lacking *Acvr1* expression in *Col2a1+* cells show a global delay in chondrogenesis and a delay in osteogenesis in the digits [12]. Mice lacking *Acvr1* expression in *Prrx1+* osteochondroprogenitor cells show significant disruption of the first digit, with soft tissue connecting two proximal elements to an unidentified distal element [68]. Mice expressing the R206H mutation only in *Prrx1+* cells have severe limb deformities [63,64] but show a particular reduction in skeletal elements, dysregulated growth plate polarity, and the loss of joint structures in the first digit of the hindlimb [67], closely mirroring the human FOP toe phenotype. Importantly, these mice also show malformations of the fifth digit, which is seen frequently in the hands of humans with FOP [8,69], but not the feet [60,69], suggesting a more general mechanism of action in the limb than a first-digit-specific effect. In mice surviving to four weeks of age, articular cartilage of the knees is significantly reduced. Though mice globally expressing the R206H mutation survive only a few hours after birth, they show a reduction in all limb skeletal elements, with the strongest effects on preaxial elements, i.e., the radius, fibula, and great toe [67]. These phenotypes were not present in mice expressing the mutation in only *Scx+* (tenocytes) or *Mx+* (bone-marrow-derived endochondral progenitors) cells [70]. Mice expressing the engineered, constitutively active mutant receptor Acvr1-Q207D typically do not survive gestation, though mice expressing that receptor only in *Nfatc1+* cells showed HO in the wrists, ankles, and phalanges in association with joint tissues, as opposed to originating from within the skeletal muscle tissue [71]. The infection of chicken embryos with retrovirus containing the R206H, Q207D, or Q207E mutant receptors leads to fused joints in the limbs and digits [40], suggesting an evolutionarily conserved function for *Acvr1* in joint and digit development. Together, these studies show that Acvr1 acts during skeletal patterning to regulate chondrogenesis, as well as the site-specific contributions of cells to joint structures.

*Acvr1* clearly plays important roles in early skeletal patterning, particularly in the limbs, but the exact molecular mechanism or mechanisms of this contribution and how it achieves such remarkable specificity are unknown. In mammals, the first digit is transcriptionally distinct from the other four in that it is considered *SHH*-independent and only expresses one of the 5′ *HOXD* cluster indispensable for digit morphogenesis [72]. However, digit malformations in both humans and mice with FOP-causing mutations are not restricted to digit one. While this does not preclude a role for the first digit’s unique identity in the etiology of the FOP great toe phenotype, it suggests a more broadly acting mechanism. For instance, one role of BMP in the developing limb is to ultimately interrupt cell proliferation and drive differentiation; therefore, BMP activity is at first inhibited in the most distal cells to permit continued outgrowth. The disruption of that inhibition leads to premature chondrification and reduced digit length [73]. Mice expressing Acvr1-R206H show significantly reduced digit length and possible altered SMAD1/5/(8/9) in the distal digit tips, though the latter observation has not been quantified [67]. Another possible explanation lies in the molecular mechanism of determining the number, position, and size of skeletal elements in the hands and feet. These features are thought to be partially determined by Türing reaction–diffusion, in which a long-range initiator of a signal and a short-range inhibitor of that signal interact with local geometry over time to determine stable active and inactive zones [61]. BMP pathway activity and Sox9-driven chondrogenesis are thought to be positive/active outputs of this pathway, with regions of high BMP pathway activity driving the differentiation of the cartilage template for the skeleton, but the specific components and the results of small perturbations are not fully understood. Therefore, it may be that the altered responsiveness of the Acvr1-R206H receptor to stimuli and leaky signaling disrupt the normal generation and maintenance of this reaction–diffusion mechanism in such a way that only specific digits are affected. Variations in either BMP-regulated limb bud outgrowth or a BMP component of Türing reaction–diffusion should therefore produce a gradient of severities. Though a clean gradient of phenotypes has not been described, variant FOP-causing mutations can lead to a variety of distal limb phenotypes, including syndactyly, the loss of digits, the loss of all medial phalanges, and the loss of nail beds (i.e., disruption to the distal tips of digits) [38].

## 4. Altered Skeletal and Joint Development in FOP

The developmental skeletal phenotype of FOP appears to be primarily due to disrupted joint formation. Understanding the specifics of toe malformation can help to understand both its etiology and how it may be connected to other, less common developmental features of FOP (Figure 2). These include hip dysplasia [74,75], hand malformations [69,74,76], fusions of the cervical spine [76,77], distinct craniofacial features [78,79,80], fusions of the temporomandibular joint [79], metaphyseal osteochondromas [81], and exostosis-like mineralized tendon insertion sites [82]. Skeletal malformations in FOP must be distinguished between primary malformations caused by altered development and secondary deformities resulting from the presence of heterotopic bone or altered posture and gait [74,83]. In particular, the tibio-fibular joints, femoral necks, and cervical spine show fusion and dysmorphia in early life, which, in the absence of HO near the affected areas, suggests those symptoms have developmental origins [84]. Large-cohort natural history studies support that osteochondromas, osteophytes, fusions of spinal elements, joint degeneration, hip dysplasia, and intra-articular ankyloses of costovertebral joints may also occur in the absence of focal HO, suggesting that the FOP mutation drives degenerative symptoms separate from HO well after primary skeletal development has completed [74,84,85]. Pertinent to laboratory studies, genetic knock-in mouse models are true to these features of FOP: the global or mosaic knock-in of Acvr1-R206H can drive all of the aforementioned skeletal phenotypes [63]; *Prrx1*-specific expression can drive all limb phenotypes [64]; and the *Sox10*-specific expression of Acvr1-R206H or the *P0*-specific expression of Acvr1-Q207D can drive craniofacial phenotypes, the latter due to altered cranial neural crest cell migration [86,87]. Notably, the appearance of osteochondromas, intra-articular ankyloses, the exostosis-like mineralization of tendons, and osteophytes are all disorders involving the aberrant growth of bony tissue not necessarily in the context of injury. These suggest a more general ability of the FOP mutation to drive osteogenesis in multiple clinically relevant contexts, rather than only during flare-ups. While the precise impact of each of these phenotypes is uncertain, the non-HO characteristics of FOP contribute to a gradual loss of mobility, and those affecting the costovertebral joints are implicated in thoracic insufficiency syndrome, which is responsible for a high percentage of morbidity in FOP [88].

We will first consider the events of embryonic joint development before applying these concepts to that of altered digit and joint patterning in FOP (Figure 1). On a cellular level, presumptive skeletal joints are identified by low canonical BMP pathway activity, the expression of *Gdf5* (a BMP pathway ligand), and the gradual migration of *Gdf5+* cells into the joint interzone, the space directly between two adjacent bones (Figure 1) [89,90,91]. These cells are specified based on the timing of this migration, with early-migrating cells differentiating into tissues, including articular cartilage, and late-migrating cells contributing to tendons, ligaments, and synovium [92]. The disruption of *Gdf5+* cell migration can lead to site-specific defects ranging from a failure of joint cavitation and interzone formation to a delayed or aberrant development of tendons, ligaments, and articular cartilage. The mutation of *GDF5* is associated with multiple brachydactylies (OMIM #615072, #112600, and #113100) and appendicular dysplasias (OMIM #200700 and #228900). BMP must be inhibited at sites of joint formation to prevent the differentiation of cells to cartilage and to allow the joint to cavitate [20,21]. The Wnt pathway signaling, which often acts in opposition to the BMP pathway, must be active in the interzone [93]. The genetic deletion of BMP receptors [94] or members of the Wnt pathway [93] can lead to a failure of joint progenitors to migrate and differentiate in a site-specific manner. Activins, TGF-ß, and their associated receptors are present in the developing limb, and TGF-ß pathway signaling activity is required for skeletal development. However, the precise roles of these factors at the level of digit joint formation have not yet been untangled (reviewed in depth elsewhere [95]). The loss of *Noggin*, a primary BMP ligand antagonist, leads to a total loss of synovial joint formation in mice [25] and a spectrum of digit phenotypes in humans, including brachydactyly type B2 (OMIM #611377) and proximal symphalangism 1A (OMIM #185800). Thus, while the differential contributions of receptors, ligands, and resultant complexes may have joint-specific roles, BMP inhibition is absolutely required for proper joint development.

In situ hybridization experiments in *Acvr1^R206H/+^; Prrx1-Cre* mouse limbs suggest that *Gdf5+* cells do not properly localize to the presumptive joint interzone in a digit-specific manner [67]. This altered localization is concurrent with aberrant pSMAD1/5 activity in the presumptive digit joint interzones and directly precedes defective skeletal and joint patterning in the digit. However, other joints develop mostly normally, matching observations in the FOP human patient population, suggesting the presentation is more complex than a generic loss of BMP pathway inhibition in all joints, as in *Noggin* knock-out models. The FOP mouse phenotype is more similar to *Gdf5* homozygous knock-out [96], and an FOP-like great toe malformation has been reported in one patient deficient for *BMPR1B* [97] and another with a potentially causative point mutation in *BMPR1B* [98]. *BMPR1B* is a type I receptor that preferentially binds GDF5 ligands to drive joint formation, the loss of which is associated with human brachydactylies (OMIM #616849 and 112600) [99], suggesting that the FOP joint phenotype may be primarily, though not entirely, due to a disruption in GDF5 function. An intriguing possible explanation for this is that even though GDF5 can act as a context-specific chondrogenic factor by activating the canonical BMP pathway, it may instead act as a high-binding, low-activity sink for receptors in the joint interzone [100], an idea supported by GDF5’s typically weak activation across different receptors [22]; therefore, the constitutive activation of ACVR1 by the R206H mutation may circumvent such a function, leading to joint fusion in a pattern remarkably similar to GDF5 loss of function. In the brachydactylies referenced above, axial joints are typically unaffected, while appendicular joints are highly affected. Thus, the development of joints other than those in the digits may progress without defects, or with less severe defects that then become more apparent when challenged (reviewed elsewhere [101,102]), possibly explaining the early degenerative joint disease observed in FOP patients.

Changes in signaling in adult joint tissues that disrupt tissue maintenance and homeostasis can also presage degenerative osteoarthritis (OA), the appearance of osteochondromas, and the mineralization of cartilaginous connective tissues [103], all of which are observed at high rates and early ages in FOP compared to the general population [84]. The balance of BMP and TGF-ß signaling is critical in articular cartilage maintenance. One primary factor in OA is chondrocyte hypertrophy, a process normally driven by BMP pathway signaling during embryonic development. A mouse model of osteoarthritis shows the upregulation of the TGF-ß receptor Cripto, which participates in a BMP-pathway-activating complex [104]. The inhibition of Acvr1-BMP signaling has successfully reduced osteoarthritis progression in mice [105]. ACVR1 was also up-regulated in hypertrophic chondrocytes taken from the articular cartilage of osteoarthritis patients [106]. Together, these implicate enhanced basal BMP signaling as a candidate for the joint degeneration seen in people with FOP.

The localization of joint pathology in FOP suggests several factors acting individually or in concert at different sites in the body. One is altered interactions with locally expressed genes such as *GDF5* during embryonic development and early childhood, thus mimicking the joint diseases usually associated with these genes. A second factor is minor alterations in embryonic and childhood joint development that are not initially observable but become apparent as the individual ages. A third is altered BMP pathway signaling within the mature joint that leads to premature joint degeneration. Finally, there are HO-caused changes in posture and gait, which can damage articular cartilage and other joint structures over time. Considering the variability in disease progression among individuals, it is likely that these combine uniquely depending on natural history.

## 5. Non-Skeletal Symptoms of FOP

Though FOP is primarily a disease of HO formation and altered skeletal development, other symptoms also arise with variable penetrance, especially in rare, variant cases in which ACVR1 mutations other than R206H are observed. These include alopecia, the loss of fingernails and toenails, and severe conductive hearing loss [38], as well as a suite of cardiopulmonary and neurologic phenotypes reviewed by Khan et al. [107]. The involvement of multiple organs apart from the skeletal system is in some ways to be expected because the BMP pathway is evolutionarily ancient and plays critical roles in morphogenesis throughout the body. Though specific molecular developmental mechanisms for the below listed phenotypes have not been thoroughly investigated, we may speculate on how ACVR1-R206H is able to manifest them.

Alopecia: FOP patients with either classic or variant mutations may present with thinning or lost hair. The molecular development and maintenance of hair follicles relies on gene networks including BMP and Wnt signaling, with BMP promoting a quiescent state and Wnt promoting an active one [108]. The loss of hair-follicle-specific BMP signaling results in dysfunctional follicular morphogenesis [109]. The overexpression of *Acvr1* in the hair follicle alters follicle morphology and localization, as well as wound healing [110]. Thus, hair thinning and hair loss in FOP are likely a primary effect of the hyperactivation of BMP pathway activity by ACVR1-R206H, though the effects of this specific mutation in this context have not been rigorously investigated.Loss of fingernails and toenails: Variant FOP mutations may also be associated with the loss of some or all fingernails and/or toenails. The nail bed is a densely cellular tissue with an active stem cell niche maintained by Wnt pathway signaling, whereas BMP is implicated in terminal nail cell differentiation [111]. The aberrant activation of BMP pathway signaling in the nail bed may deplete this stem cell population or prevent it from forming during early development. Further, nailbeds are features of the distal tip of the digits, and if digit patterning is so severely altered by pathogenic ACVR1 signaling as to stunt digit outgrowth, a niche for the nailbed might never be established.Hearing loss: The pathogenesis of hearing loss in FOP is symptomatically mostly conductive due to developmental fusion of the ossicles of the middle ear. However, some individuals with FOP have neural hearing impairment. The cochlea develops primarily under the control of Wnts, Fgfs, and Shh, but also requires a gradient of BMP pathway activity to refine sensory structures and aide in the development of hair cells [112]. While the expression of *ACVR1* in sensory organs has not been detailed, the disruption of this gradient may lead to impaired neural hearing in some individuals with FOP.Cardiac phenotypes: Individuals with FOP have long been observed to have subclinical cardiac anomalies observed by electrocardiogram [113,114]. ACVR1 function is required for the development of multiple components of the heart such as the endocardial cushion [115,116]. Because of the known roles of ACVR1 in heart development, it is possible cardiac anomalies are primary symptoms of altered BMP pathway signaling; however, FOP also frequently restricts the chest wall, which may lead to changes in cardiopulmonary function as well.Neurologic dysfunctions: FOP is associated with a range of neurological symptoms [117]. Neuropathic pain, focal demyelination, and central nervous system patterning in general have all been linked to BMP pathway signaling. Mice expressing Acvr1-R206H have significant focal demyelination, and a clinical report showed multiple demyelination lesions in four FOP patients, though the demyelination could not be directly linked to neuropathy [118]. While the specific molecular mechanism has not been investigated, there are clear avenues for ACVR1 mutations to lead to multiple neurological defects.

## 6. FOP Treatments and Skeletal Development

Recently, several therapeutic approaches for treating FOP have shown promise in the prevention and abrogation of flare-ups and subsequent HO. While HO is the primary symptom contributing to reduced quality of life, other manifestations of the disease discussed here prompt a consideration of how treatments for FOP may impact the progression of phenotypes of the normotopic skeleton and joints. Clinical trials of FOP provide several unique challenges which have been thoroughly outlined [119]. Though HO is preferentially confirmed by computed tomography (CT), acquiring these data requires patients to assume positions that may be painful, dangerous, or impossible due to immobilizing HO. Such limitation also impacts the collection of data on joint and normotopic skeletal health. Therefore, patient- and clinician-reported outcomes must often be relied upon. In designing and assessing future clinical trials and the long-term outcomes of potential therapeutic agents, it will therefore be important to assess joint health and the potential retardation of developmental arthropathy as a viable clinical outcome independently of HO. Until then, investigators and clinicians, including the majority of trials referenced below, make use of the cumulative analogue joint involvement score (CAJIS). CAJIS was developed to be a snapshot of mobility burden in patients with FOP and assesses the impact of HO on joint function, but also provides inference into possible joint deterioration in the absence of reported or observed flare-ups and HO [120].

The following therapies are in recently completed, ongoing, or upcoming clinical trials and are detailed on the International FOP Association website (https://www.ifopa.org/ongoing_clinical_trials_in_fop; accessed on 24 June 2024) as of the publication of this article. While a thorough examination is beyond the scope of this review, we provide a brief consideration of how each relates to non-HO symptoms of FOP.

Palovarotene is a highly specific retinoic acid receptor gamma (RARγ) agonist that successfully abrogates cardiotoxin injury-induced HO in mouse models of FOP and appears to improve growth plate health, which is compromised by the Acvr1 R206H mutation [64,121]. However, concerns were raised early on about the skeletotoxicity of palovarotene in mouse models, which caused synovial joint hypertrophy followed by the premature closure of growth plates, leading to reduced growth and development [122]. This may be due to the off-target effects of the anti-chondrogenic properties of RAR agonists, which may deplete the stem cell pools in growth plates needed to continue longitudinal bone growth, and may implicate a delicate balance among Acvr1, RARγ, and other signaling pathways in growing bone. In clinical trials, the drug reduced both flare-up incidence and the progression of HO in patients but had side-effects including the premature closure of growth plates [123]. An upcoming long-term palovarotene study (NCT06089616) will include skeletal age, physeal closure, and height velocity as secondary outcomes, which may provide some insight into non-HO symptoms, though they may be conflated by HO lesions impinging on growth and joint function.

Rapamycin is a well-known inhibitor of mTOR kinase with immunosuppressant functions currently in clinical trials to treat FOP (UMIN000028429). As mentioned previously, mTOR and its regulation of HIF1α are potent targets for preventing traumatic and genetic HO [59], as well as their involvement in osteoarthritis and other degenerative joint diseases [124]. It will therefore be especially interesting to note whether rapamycin has any effect on non-HO-related changes to CAJIS.

Garetosmab, an activin A-blocking antibody, entered clinical trials [125] following successful reduction in HO in mouse models [49]. This treatment was not reported to have deleterious developmental defects; however, the mouse studies did not investigate the effects on growth plate closure or synovial joints. The current trial (NCT05394116) includes an assessment of patient joint function that may allude to degenerative joint disease but, like other trials, does not include specific indicators of joint disease as study parameters.

Recent investigations in the clinic and in FOP mouse models revealed that even the partial inhibition of matrix metalloproteinase-9 (MMP-9) activity by genetic, pharmacologic, or biologic means potently inhibits HO. Thus, it appears that MMP-9 is a vital molecular link between inflammation and HO in FOP, unveiling a novel treatment strategy for FOP [126,127].

A primary difficulty with pharmaceutical options for FOP has been the close sequence similarity between ACVR1 and other type I receptors, which creates high risk for side effects when ACVR1 is directly targeted. Modern small-molecule inhibitors either designed to or able to minimize off-target effects are now in various stages of clinical trials as well, including zilurgisertib (INCB000928), fidrisertib (IPN60130), and saracatinib (AZD0530). Due to their high specificity, these therapies may show promise for mitigating both HO and non-HO symptoms.

Adeno-associated virus (AAV)-based therapies have soared in use and pursuit in the past several years. Recently, Yang and colleagues published a study using AAV to suppress transcripts encoding ACVR1-R206H and produce healthy *ACVR1* transcripts [128,129]. Encouragingly, this method showed decreased HO load in mice, as well as reductions in degenerative joint disease symptoms, vertebral fusions, and osteochondromas [128]. Though AAV therapies for FOP have not yet reached clinical trials, these preliminary investigations show exciting promise.

While HO correctly remains the primary target of potential FOP therapies, the growth plate health and joint tissue phenotypes that worsen with age represent a knowledge gap that should be considered during the assessment of clinical trial outcomes as new therapies are investigated. The nature of the disease precludes or makes onerous certain definitive measures of joint and skeletal health, however, so there is understandably a practical limit to the acquisition of such data.

## 7. Conclusions


*“To measure the great toe of the foot is to measure the giant.”*
Victor Hugo

FOP is a complex disease despite its deceptively simple genetic origin: a single, recurrent base-pair substitution leading to a devastating outcome. While the burden of HO is certainly the most critical component of the disease to be addressed, our understanding of the molecular etiology of other symptoms associated with FOP has greatly improved in the past several years. Understanding the cellular and molecular signaling in a disease and looking deeply into seemingly minor phenotypes can yield a vastly improved comprehension of both fundamental biological processes and those of rare diseases, with significant improvements in quality of life. FOP is a disease not just of HO formation, but also of joint, skeletal, cardiac, and neurological development and maintenance. As exciting new therapies are developed and tested, it is important to understand the many different temporal and biological niches of BMP signaling to give patients the best possible information and outcomes.

## Figures and Tables

**Figure 1 biomolecules-14-01009-f001:**
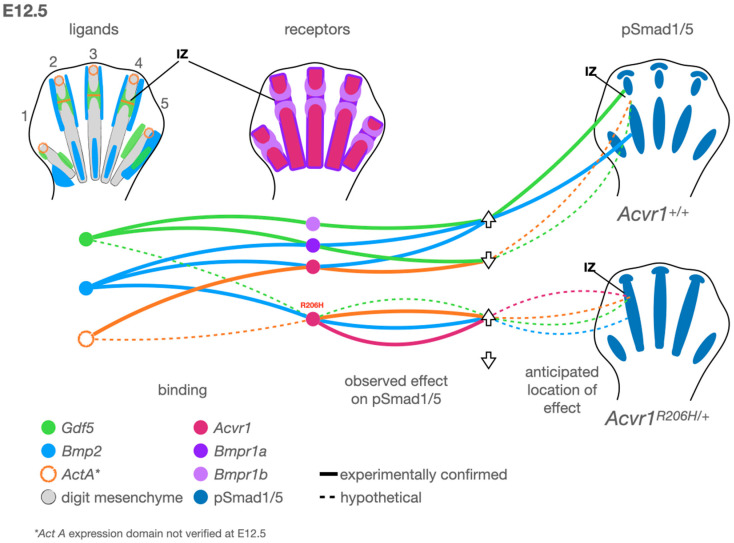
A model for altered regulation of BMP pathway activity during digit development by Acvr1 R206H. **Left**: Observed expression patterns of the ligands Gdf5 (green), Bmp2 (light blue), and ActA (orange, based on data from chickens). Middle: Type 1 receptors Acvr1 (fuchsia), Bmpr1a (purple), and Bmpr1b (light purple). **Right**: Observed pSmad1/5 domains (dark blue) in control (**top right**) and Acvr1^R206H/+^ (**bottom right**) mouse limbs. At this early stage, digits 1 and 5 have not yet acquired the transverse stripes of *Gdf5* expression associated with joint interzones (IZ). Arrows indicate whether the ligand and receptor connected by the associated line up- or down-regulates canonical BMP pathway activity via Smad1/5 phosphorylation. Solid lines indicate binding and activity observed experimentally. Dotted lines indicate hypotheses supported by altered Acvr1 activity in the presence of FOP-causing mutations. Gdf5 attenuates BMP-pSmad1/5 activity through Bmpr1a binding in a context-dependent manner, which is hypothesized (dotted line) to restrict BMP pathway activity at the interzone (IZ). In vitro, Bmp2-Acvr1 binding activates pSmad1/5, but ActA-Acvr1 binding reduces or nullifies pSmad1/5. Acvr1-R206H binds Bmp2 as a potent activator and cells expressing Acvr1-R206H strongly activate pSmad1/5 when treated with ActA. Acvr1-R206H can also signal ligand-independently. Based on expression patterns, receptor–ligand binding capabilities, and signaling through these complexes, we hypothesize the following: ligand-independent Acvr1-R206H activity contributes to multiple aspects of the observed phenotype; ActA aberrantly promotes IZ pSmad1/5 activity in a digit-independent manner; Gdf5-Acvr1 signaling promotes digit-specific IZ pSmad1/5 activity; and Bmp2-Acvr1-R206H likely contributes to persistent chondrogenic activity in the developing digit skeleton.

**Figure 2 biomolecules-14-01009-f002:**
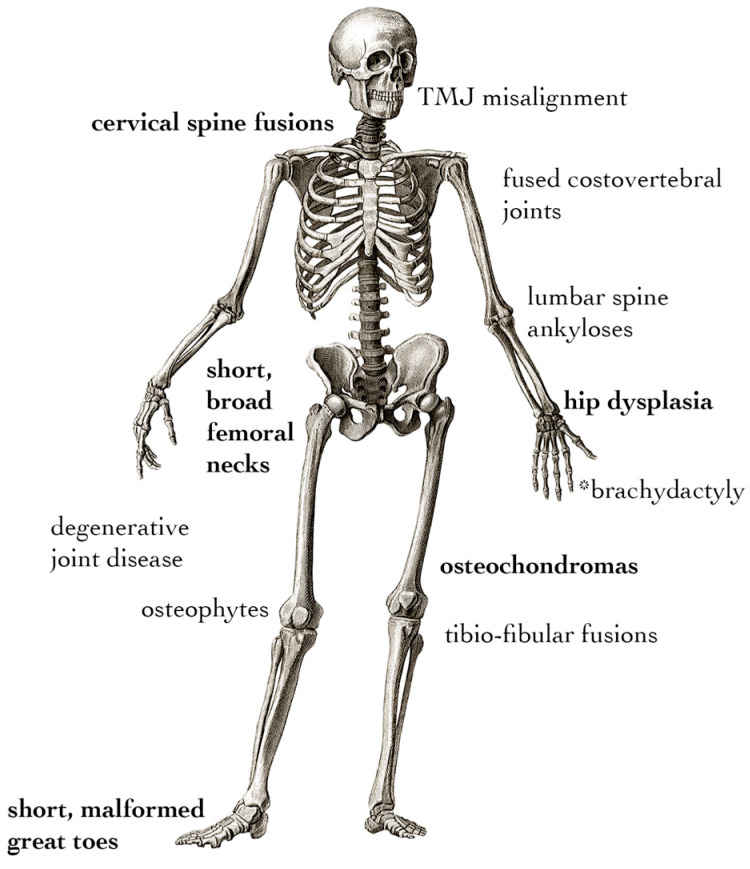
Non-HO skeletal phenotypes of FOP. Symptoms in **bold** are present in the majority of individuals with FOP, whereas the remainder arise with variable penetration among patients. * Brachydactyly is only associated with non-R206H variant cases.

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
