# Peer review of "Molecular Developmental Biology of Fibrodysplasia Ossificans Progressiva: Measuring the Giant by Its Toe"

_biomolecules, 2024, doi:10.3390/biom14081009_

Round 1

Reviewer 1 Report

Comments and Suggestions for Authors

In the paper on FOP by leading authors in the field this devastating disease is extensively described both morphologically, pathologically and mechanistically. All relevant recent publications have been included, as well as a short presentation of clinical trials with developed therapies. I would suggest to extend the portion 6. with other ongoing clinical studies presented at the disclosed FOP website to help investigators not familiar with the subject to understand all FOP therapeutic targets developed by various institutions, mainly in the US and Japan, with an option of short explanation of protocol design due to inability of majority of patients to enter NMR and CT machines at variable intervals following therapy and main primary and secondary outcomes of such very difficult to conduct clinical trials. 

Regarding Figure 1, it is just a demonstration of the skeleton, but several soft tissue organs have been named as targets for pathological changes (demyelination, alopecia, cardiac abnormalities, neuropathic pain, etc. which all do not fit into the figure of a regular skeleton). 

Author Response

Comment 1: "I would suggest to extend the portion 6. with other ongoing clinical studies presented at the disclosed FOP website to help investigators not familiar with the subject to understand all FOP therapeutic targets developed by various institutions, mainly in the US and Japan, with an option of short explanation of protocol design due to inability of majority of patients to enter NMR and CT machines at variable intervals following therapy and main primary and secondary outcomes of such very difficult to conduct clinical trials."
Response 1: We thank the reviewer for this suggestion. We have amended Section 6 to include all trials currently on the referenced site as well as a brief discussion of the challenges associated with acquiring trial outcomes. The rapamycin and MMP9-related therapies are now included. We felt this change necessitated a brief mention of mTOR at the end of Section 2 and have amended the text accordingly.
Comment 2: Regarding Figure 1, it is just a demonstration of the skeleton, but several soft tissue organs have been named as targets for pathological changes

Response 2: We agree with the reviewer that the figure can be improved. We have removed soft-tissue phenotypes from the image because those are discussed separately and only briefly, and chosen instead to use the figure to indicate non-HO, skeletal phenotypes. The figure legend has been updated accordingly.

Reviewer 2 Report

Comments and Suggestions for Authors

Fibrodysplasia ossificans progressiva (FOP) is a rare genetic connective tissue disorder characterized by heterotopic ossification (HO). The manuscript authored by Towler et al., titled " Molecular Developmental Biology of Fibrodysplasia Ossificans Progressiva: Measuring the Giant by its Toe," focuses on the molecular mechanisms underlying the developmental phenotypes of FOP, particularly the role of ACVR1 in skeletal patterning and growth. Additionally, the review emphasizes the importance of advancing patient care, assessment, and treatment, as well as contributions to the fields of bone and joint biology through better understanding the disease. This review is thoughtfully designed and well-organized, making it a suitable candidate for publication in the journal Biomolecules.

Comments on the Quality of English Language

1.         On page 2 under the title of “2. Molecular Basis of FOP”, the word “The” should be deleted.

Author Response

Comments 1: On page 2 under the title of “2. Molecular Basis of FOP”, the word “The” should be deleted.

Response 1: We thank the reviewer for their kind summary remarks and encouraging additional passes for grammatical errors. This error and a small number of others have been amended in the text and highlighted.

Reviewer 3 Report

Comments and Suggestions for Authors

This is an informative review article, primarily focused on the plausible mechanisms by which developmental skeletal malformations occur in FOP, and other effects of the FOP mutations not related to heterotopic ossification.  The authors have nicely laid out these two different scenarios, and have cited appropriate literature.  There are just a few comments as listed below:

1.    Page 2, paragraph 2 – “mesenchymal stem cells” – This is not an accurate to use under many circumstances.  During development, mesenchyme gives rise to connective tissues, blood vessels and blood.  But in the context described in this review by the authors, the cells are more committed than that, and could be called skeletal progenitor cells, or something to that effect.  While there are stem cells in the population that gives rise to the two connective tissues of interest in this article (cartilage and bone), not all of the cells are stem cells, and they should not be generalized as “mesenchymal stem cells.” 

2.    Page 4, paragraph 2 “induced mesenchymal stem cells…”. Irrespective of how one uses the “mesenchymal (stem) cells” term during development, the use of the “MSC” term is not appropriate here.  Once does not establish a culture of “MSCs” in vitro.  Only a small subset of cells within that population are true stem cells, while the remainder are transiently amplifying cells and more committed cells.  In these situations, the authors should more accurately describe the cells by indicating their source.  In fact, looking at the original article (reference 37), the authors used induced pluripotent stem cells, and made what they called “induced mesenchymal stromal cells.”  However, to date, the proof that people have actually made functional “MSCs” from iPSCs is slim, and has relied on expression of a few fibroblastic cell surface markers that do not predict either stemness or functionality of the cells.  The differentiation capacity by assays such as in vivo transplantation are needed.  In reference 37, the authors showed that pellet cultures made from mutant “iMSCs” showed signs of mineralization, but did not show actual bone formation.  Of note, dystrophic calcification is a form of HO, but is not FOP.  These studies need to be taken with a grain of salt with respect to the consequences of the FOP mutations on the function of skeletal cells. 

3.    Page 6, paragraph 1 – Can the authors expand on the sentence “with regions of high BMP pathway activity pre-patterning the cartilage template for the skeleton…” Mechanistically, what exactly is pre-patterning?  In the last sentence, the authors most likely mean “disruption of the distal tips of digits.” 

Author Response

Comment 1: Page 2, paragraph 2 – “mesenchymal stem cells” – This is not an accurate to use under many circumstances.  During development, mesenchyme gives rise to connective tissues, blood vessels and blood.  But in the context described in this review by the authors, the cells are more committed than that, and could be called skeletal progenitor cells, or something to that effect.  While there are stem cells in the population that gives rise to the two connective tissues of interest in this article (cartilage and bone), not all of the cells are stem cells, and they should not be generalized as “mesenchymal stem cells.” 

Response 1: We appreciate the reviewer’s thoughtful consideration of the specific language and agree. The text has been amended to “skeletal progenitor” where indicated.

Comment 2: Page 4, paragraph 2 “induced mesenchymal stem cells…”. Irrespective of how one uses the “mesenchymal (stem) cells” term during development, the use of the “MSC” term is not appropriate here.  Once does not establish a culture of “MSCs” in vitro.  Only a small subset of cells within that population are true stem cells, while the remainder are transiently amplifying cells and more committed cells.  In these situations, the authors should more accurately describe the cells by indicating their source.  In fact, looking at the original article (reference 37), the authors used induced pluripotent stem cells, and made what they called “induced mesenchymal stromal cells.”  However, to date, the proof that people have actually made functional “MSCs” from iPSCs is slim, and has relied on expression of a few fibroblastic cell surface markers that do not predict either stemness or functionality of the cells.  The differentiation capacity by assays such as in vivo transplantation are needed.  In reference 37, the authors showed that pellet cultures made from mutant “iMSCs” showed signs of mineralization, but did not show actual bone formation.  Of note, dystrophic calcification is a form of HO, but is not FOP.  These studies need to be taken with a grain of salt with respect to the consequences of the FOP mutations on the function of skeletal cells. 

Response 2: We appreciate the reviewer’s thoughtful consideration of this terminology. We have accordingly changed the text to read “induced pluripotent stem cells.”

Comment 3: Page 6, paragraph 1 – Can the authors expand on the sentence “with regions of high BMP pathway activity pre-patterning the cartilage template for the skeleton…” Mechanistically, what exactly is pre-patterning?  In the last sentence, the authors most likely mean “disruption of the distal tips of digits.” 

Response 3: “Pre-patterning” is here used to indicate molecular activity occurring before and during the consequent differentiation of cells caused by that activity. That is, the pattern of that activity predicts and matches the subsequent pattern of cartilage. However, we appreciate that without having previously used or defined that terminology in this manuscript, that may be unclear, and have thus amended the text to say, “ … high BMP pathway activity driving differentiation of the cartilage template for the skeleton.”